# Reporter Replicons for Antiviral Drug Discovery against Positive Single-Stranded RNA Viruses

**DOI:** 10.3390/v12060598

**Published:** 2020-05-30

**Authors:** Rafaela S. Fernandes, Marjorie C. L. C. Freire, Renata V. Bueno, Andre S. Godoy, Laura H. V. G. Gil, Glaucius Oliva

**Affiliations:** 1Physics Institute of São Carlos, University of São Paulo, São Carlos 13566-590, SP, Brazil; rafaela.fernandes@usp.br (R.S.F.); marjorie_freire_@hotmail.com (M.C.L.C.F.); buenorvieira@gmail.com (R.V.B.); andre_godoy@yahoo.com.br (A.S.G.); 2Instituto Aggeu Magalhães, IAM-FIOCRUZ, Recife 50670-420, PE, Brazil; lgilfiocruz@gmail.com

**Keywords:** replicon systems, (+) ssRNA viruses, direct-acting antivirals, drug discovery

## Abstract

Single-stranded positive RNA ((+) ssRNA) viruses include several important human pathogens. Some members are responsible for large outbreaks, such as Zika virus, West Nile virus, SARS-CoV, and SARS-CoV-2, while others are endemic, causing an enormous global health burden. Since vaccines or specific treatments are not available for most viral infections, the discovery of direct-acting antivirals (DAA) is an urgent need. Still, the low-throughput nature of and biosafety concerns related to traditional antiviral assays hinders the discovery of new inhibitors. With the advances of reverse genetics, reporter replicon systems have become an alternative tool for the screening of DAAs. Herein, we review decades of the use of (+) ssRNA viruses replicon systems for the discovery of antiviral agents. We summarize different strategies used to develop those systems, as well as highlight some of the most promising inhibitors identified by the method. Despite the genetic alterations introduced, reporter replicons have been shown to be reliable systems for screening and identification of viral replication inhibitors and, therefore, an important tool for the discovery of new DAAs.

## 1. Introduction

Single-stranded positive RNA ((+) ssRNA) viruses include a group of human pathogens with significant socioeconomic impacts, such as Dengue virus (DENV), West Nile virus (WNV), Zika virus (ZIKV), Hepatitis C virus (HCV), Severe Acute Respiratory Syndrome Coronavirus (SARS-CoV), Middle East Respiratory Syndrome Coronavirus (MERS-CoV), Severe Acute Respiratory Syndrome Coronavirus 2 (SARS-CoV-2), Poliovirus (PV), Enterovirus 71 (EV-71), Hepatitis E virus (HEV), and many others. Some members are responsible for causing significant outbreaks worldwide, as exemplified by the ZIKV pandemic 2016–2017 [1], while others, such as DENV, PV, and HEV, for example, are endemic in many tropical countries, representing a constant threat for large population groups [2]. Despite recent advances in the field of drug discovery, most of these infections remain without a satisfactory treatment [3,4]. The emergence of the new coronavirus (SARS-CoV-2) in China in 2019, which rapidly developed into a global pandemic health crisis [5,6], highlights the urgent need for effective and specific new antiviral drugs against (+) ssRNA viruses.

Due to their low-throughput nature, traditional antiviral assays, such as cytopathic effect evaluation (CPE) and viral-antigen based assays, are not ideal for testing large compound libraries [7,8,9]. In addition, the required biosafety level (BSL) of the facilities to manipulate viral pathogens, as most of them are classified as BSL-3 agents, is also a limiting factor for the discovery of new inhibitors. Therefore, antiviral development requires reliable biological assays that can be performed in a high-throughput screening (HTS) format, such as replicon-based assays [10,11].

Replicons are self-replicative subgenomic systems in which the genes encoding viral structural proteins are replaced by a reporter gene. Since mutations can only be introduced into DNA, the genomes of RNA viruses must first be reverse transcribed. The resulting complementary DNA (cDNA) can be mutagenized, generating non-infectious replicons that possess all genetic elements necessary for self-replication, but lack the structural genes responsible for the production of progeny viruses [11,12,13]. The correct 5′ and 3′ ends of the viral genome are crucial to the success of replicon constructs, as many viruses have specific structures at their terminals for replication and/or translation, such as a cap or a covalently linked virus-encoded peptide VPg at the 5′ end, can also form loop structures or have a poly-A tail in the 3′ end. A single restriction enzyme site inserted into the plasmid downstream of the 3′ end of the cDNA allows linearization before run-off in vitro transcription, resulting in a 3′ end of the transcribed RNA identical or as similar as possible to that of the viral genome [11,12]. In relation to the 5′ end, the use of bacteriophage promoters, such as T7 or SP6, allows RNA transcription with a marginally modified or even the desired start, since only a 5′ G residue is necessary for efficient transcription by these enzymes [11,12]. Alternatively, the replicon sequence can be cloned under the control of eukaryotic promoters, such as the cytomegalovirus (CMV) promoter. In this case, transcription of the subgenomic RNA occurs in the cellular nucleus, and production of the authentic 3′ end is mediated by a hepatitis delta virus ribozyme (HDVr) cloned downstream of the genome. The establishment of replicon constructs driven by a CMV promoter allows for direct transfection of plasmid DNA [11,14].

Generally, a reporter gene is introduced in place of the deleted structural genes to allow luminometric (luciferases) or fluorescent (fluorescent proteins) detection of RNA replication [11,14]. To release the reporter protein, a foot-and-mouth disease virus (FMDV) 2A autoprotease is inserted downstream of the reporter gene [11,14]. Alternatively, the introduction of the reporter gene can be achieved by constructing bicistronic replicons with a stop codon included at the 3′ terminus of the insert. The reporter protein is still expressed from the 5′ terminal region, and translation initiation of the nonstructural proteins is then mediated by an internal ribosomal entry site (IRES) derived from the encephalomyocarditis virus (ECMV). An IRES-driven reporter gene cassette can also be inserted after the ORF into the 3′ UTR of the viral genome [11,14].

Transfection of susceptible cells with replicon systems results in the transient expression of the reporter protein, the level of activity of which would reflect the extent of viral replication for a limited period of time (Figure 1). Inhibitors of viral replication decrease the expression and activity of the reporter protein. This loss-of-signal end point can produce false-positive hits, resulting from molecules that interfere with the activity of the reporter protein and those that adversely affect cell health. For this reason, a cytotoxicity assay, performed in parallel with the primary screen, assists in the discovery of specific viral inhibitors [15]. Replicons can also be stably maintained in cells by introducing a drug resistance gene into the system, such as puromycin N-acetyltransferase (Pac) or neomycin phosphotransferase (Neo), simplifying high-throughput antiviral screenings [9].

Nevertheless, replicon-based assays allow only the discovery of molecules that affect RNA replication, but not viral entry or assembly/release. Alternatively, replicons can be packaged to produce virus replicon particles (VRPs) by providing the structural proteins *in trans* [11]. The expression of structural proteins in cells harboring the replicon RNA produces single-round infectious particles (SRIPs), which are infectious, but progeny viruses cannot propagate from the infected cells, as the packaged genome lacks structural protein genes [17]. VRPs allow screening of antiviral compounds that interfere with viral entry and/or RNA replication, where inhibition will result in reduced levels of the reporter protein [11].

The standard methods for developing replicon constructs, as well as the large genome size of some viruses, like *Coronaviridae* family members, can result in undesirable mutations or unstable/toxic clones in bacteria [12]. Therefore, the establishment of replicon systems for some viruses, such as DENV [18], WNV [19], YFV [14], HCV [20], and SARS-CoV [21], was only possible through the use of alternative hosts, low-copy plasmids, bacterial artificial chromosomes (BACs), or other improved strategies.

In this review, we compiled the recent progress made in the use of replicon-based assays for the development of antivirals against (+) ssRNA viruses, highlighting some of the most promising inhibitors identified by the method to date.

## 2. Enveloped RNA Viruses

### 2.1. Flaviviridae Family

The *Flaviviridae* is a large family of spherical-shaped viruses of ~40–60 nm in diameter, divided into four genera: *Flavivirus*, *Hepacivirus*, *Pestivirus*, and *Pegivirus*. The *Flavivirus* and *Hepacivirus* genera comprise several widely studied species, since they cause human diseases. The *Flavivirus* genome of ~11 kb encodes a single polyprotein that is co- and post-translationally processed into three structural proteins: Capsid (C), pre-membrane (prM), and envelope (E), as well as seven nonstructural (NS) proteins: NS1, NS2A, NS2B, NS3, NS4A, NS4B, and NS5 (Figure 2a). Similarly, a single polyprotein is encoded by the *Hepacivirus* genome and cleaved into four structural proteins: Core (C), envelope proteins 1 (E1) and 2 (E2), and polypeptide 7 (p7), plus six NS proteins: NS2, NS3, NS4A, NS4B, NS5A, and NS5B. Structural proteins form the viral particles and NS proteins participate in genome replication, polyprotein cleavage, modulation of cellular processes, and evasion of host immune response [22,23].

#### 2.1.1. *Flavivirus*

Infections caused by flaviviruses are either asymptomatic or result in debilitating febrile illnesses that can lead to severe symptoms, such as hemorrhage, shock, and neurological disorders [3,4]. Replicon systems for antiviral assays were successfully developed for notable members, including DENV [18,26,27], WNV [10,19,28,29], Japanese encephalitis virus (JEV) [30], YFV [14,31], and ZIKV [8,9,32].

Cell-based HTS using DENV reporter replicons have identified small-molecule compounds with antiviral activities. Using Vero (African green monkey kidney) cells stably expressing a DENV-1 reporter replicon, triaryl pyrazoline was found to inhibit the viral replication without significant cytotoxicity [26]. The inhibitory effects of two flavonoid derivatives were evaluated in BHK-21 (Baby kidney hamster) cells harboring a DENV-2 replicon expressing green fluorescent protein (GFP) and Neo; however, a potential cytotoxicity was evidenced by the low selectivity index (SI) achieved for both compounds [27]. Another BHK-21 DENV-2 replicon cell line expressing firefly luciferase (Fluc) and Neo genes was used to screen a commercially available small-molecule library of 14,400 compounds, and compound 3, a benzothiazole derivative, was able to suppress the viral replication [18].

As for DENV, subgenomic replicons were also developed for WNV and used in replicon-based assays. A WNV (New York strain—lineage I) subgenomic replicon with a large in-frame deletion of structural genes, established in BHK-21 cells, was developed by Shi et al. (2002), and proved to be capable of efficiently replicating the viral RNA [19]. Then, to develop a cell-based assay, Rluc and Neo genes were engineered into the original WNV replicon, resulting in the Rluc/NeoRep. A BHK-21 cell line expressing Rluc/NeoRep was validated as an HTS tool and used to screen a 200 small-molecule library. Compound CDDMN displayed an antiviral effect represented by an SI of >21 [10].

Replicon systems of WNV and DENV were used to confirm the antiviral effects of compounds identified as inhibitors of the NS2B-NS3 protease in a high-throughput enzymatic screening [28]. Among ~32,000 tested compounds, compound B, with a quinoline scaffold, was identified as a potent inhibitor of the NS2B-NS3 protease, possibly through the inhibition of the polyprotein cleavage, since it was also able to inhibit the viral replication of the WNV replicon [28]. Many quinoline derivatives, such as chloroquine, amodiaquine, primaquine, and others, which are Food and Drug Administration (FDA)-approved antimalarial drugs, were tested as antivirals. Amodiaquine (AQ) inhibited the replication of DENV2 Rluc reporter replicon with an EC_50_ value of 7.41 ± 1.09 µM, suggesting a possible use as antiviral agent [33]. Using the same BHK-21 WNV and DENV2 Rluc replicon cell lines developed by Mueller et al. (2008) [28], two cathecols (compound C and tolcapone) and one polyphenolic compound (tannic acid) were described as inhibitors of dengue replication in a screen performed to test ~120,000 small-molecules [29].

JEV is a major cause of viral encephalitis in Asia and is one of the only two flaviviruses (the other one being YFV) for which an effective live-attenuated vaccine is available [30,34]. Recently, a BHK-21 cell line harboring a JEV subgenomic replicon expressing Rluc and Pac selectable marker was established and demonstrated the potential to be applied for the screening of anti-JEV compounds [30]; however, no reports on the use of replicon-based antiviral assays are available to date.

Subgenomic replicon constructions based on a full-length cDNA clone of the live-attenuated 17D strain of YFV were developed by Jones and colleagues (2005). The YFV replicon expressing an Rluc gene (YF-R.luc2A-RP) [14] was used to evaluate small-molecule inhibitors of the genome replication [31]. Sindbis virus (SINV) replicon systems expressing YFV structural proteins (SIN-CprME) were used to package YF-R.luc2A-RP constructions into pseudo-infectious virus-like particles (PIVs) [14]. For an HTS assay, YFV PIVs were used to infect a monolayer of BHK-15 cells that were then treated with more than 34,000 compounds from commercial libraries. CCG-4088 and CCG-3394, naphthalene acetamide and a morpholine derivative, respectively, efficiently inhibited YFV replication with EC_50_ values of 0.4 and 1.48 µM, respectively [31].

The 2016–2017 ZIKV outbreak in the Americas has emphasized the urgency in developing HTS assays for anti-ZIKV drug discovery [9]. A full-length infectious cDNA clone of ZIKV (strain FSS13025) [32] was used by Xie et al. (2016) to develop a replicon containing Rluc and Neo genes. The reporter replicon was stablished in Huh7 (human hepatocellular carcinoma) cells and validated by testing the NITD008 inhibitor (EC_50_ value of 0.8 µM), a nucleoside analogue; however, the construct was not used in antiviral screenings [8]. More recently, Li and colleagues (2018) developed a Zika virus replicon harboring Rluc and Pac genes [9]. The ZIKV replicon was established in BHK-21 cells, validated using NITD008 (EC_50_ of 0.52 µM, and a SI > 120), and used to develop an HTS assay. The assay was optimized in a 96 well plate format and validated by testing the cytotoxicity of three antiviral inhibitors, mycophenolic acid (MPA), 2′-C-methyladenosine (2′-C-MeAdo), and 6-azauridine (6-AzUrd) [9]. All compounds displayed CC_50_ values above 12.8 µM and were also able to inhibit the luciferase activity at 2 µM. These results demonstrated that BHK21-ZIKV-Rep could be used to screen antiviral inhibitors of ZIKV replication [9].

#### 2.1.2. *Hepacivirus*

Hepatitis C virus is the major causative agent of non-A, non-B viral hepatitis, affecting 170 million people worldwide. Several distinct genetic groups of HCV are the causative agents of human hepatitis C, which is transmitted mainly by exposure to infective blood or blood products [35]. Chronic infection is the leading cause of liver disease worldwide, with about 71 million individuals at risk of developing liver cirrhosis and hepatocellular carcinoma [36]. The HCV is classified into seven recognized genotypes based on the viral genome sequence, with 67 confirmed and 20 provisional subtypes. Most HCV cases are caused by the genotype 1 (49.1%), followed by the genotypes 3 (17.9%), 4 (16.8%), and 2 (11%), all of them widely distributed around the world [37].

From the approval by the FDA in 1991 until 2014, the interferon-based therapy remained the standard treatment for chronic hepatitis. With the advent of direct-acting antiviral (DAA) drugs, the combination of pegylated-interferon-α (PEG-IFN-α) and ribavirin was gradually replaced until the end of 2017 [38]. Since 2018, the World Health Organization (WHO) has recommended the treatment of adults with chronic HCV infection with a pan-genotypic DAA combination therapy [39]. In addition to the short-term treatment, the DAA therapeutic schemes reduced the severe side effects associated with interferon-based therapy, consequently increasing the patient adherence and the treatment effectiveness [38].

It is worthy to note the pivotal role of replicon systems in discovering HCV replication inhibitors, as previously reviewed by [35,40] and more recently by [41,42]. Since the establishment of the first HCV replicon system in 1999 by Bartenschlager’s group [43], when a functional replicon was developed from a genotype 1a isolate using Huh7 cells, HCV replicons have been improved by exploring distinct permissive cell lines, virus genotypes, adaptive mutations, and their relation with drug resistance. As a consequence of the well-established HCV replicon system for HTS, promising leading inhibitors of nonstructural proteins have been identified in the past years. In this context, the main targets pursued to develop DAAs are the NS proteins NS3/NS4A protease, NS5B polymerase, and NS5A.

In addition to the enzymatic assays, cell-based systems led to the discovery of peptidomimetic NS3/NS4A protease inhibitors, with replicon models extensively used to determine barriers to resistance, genotypes’ susceptibility, and the inhibitors’ mode of action, as exemplified by the antiviral candidate BILN 2061 [44,45,46,47,48,49,50,51]. Despite being discontinued due to cardiotoxic effects detected in pre-clinical trials, BILN 2061 was a macrocyclic inhibitor important as a proof-of-concept, as it was the first of its class active in patients infected with HCV genotype I [52]. The dose–response inhibition of NS3/NS4A by BILN 2061 was determined using HCV bicistronic NS2-NS5B subgenomic replicon 1a and 1b harbored in the Huh7 cell line, with EC_50_ values of 4 and 3 nM for replicons 1a and 1b, respectively. In addition, the NS2-NS5B subgenomic replicon 1b was used to determine the BILN 2061 mechanism of inhibition by measuring its ability to block NS3-mediated polyprotein processing.

Moreover, replicon systems allowed mapping of the NS3/NS4A resistance mutations to BILN 2061, contributing to the future design of more effective antiviral drugs [53,54]. Beyond the class of macrocyclic inhibitors, linear peptidomimetic inhibitors containing an alpha-ketoamide group that covalently binds the active site were further developed, resulting in the first-wave NS3/NS4A inhibitors, Telaprevir and Boceprevir. Telaprevir and Boceprevir were the first DAAs used to treat patients infected with HCV genotype I in association with PEG-IFN-α and ribavirin, and they were approved in 2012. Telaprevir and Boceprevir benefited from HCV replicon systems, which were mainly used to characterize the anti-HCV activity and to map the drug-resistance mutations. In addition, the Telaprevir cytotoxicity, clearance, and effects in association with IFN-α were all assessed using HCV subgenomic replicons [55] corresponding to the I_377_neo/NS3-3′/wt sequence previously described by [43].

Another molecular target extensively explored during the development of anti-HCV drugs was the NS5B RNA-dependent RNA polymerase (RdRp). The NS5B RdRp inhibitors are classified as nucleoside or non-nucleoside analogues. Currently, one of each class has been approved to treat patients with chronic hepatitis C: The non-nucleoside analogue Dasabuvir and the nucleoside analogue Sofosbuvir. The NS5B RdRp enzymatic assays as well as surrogate models like the Bovine Viral Diarrhea Virus (BVDV) were used to carry out HTS towards the discovery of anti-HCV molecules. Notwithstanding, the advent of replicons has broadened the number and the chemical diversity of promising NS5B RdRp inhibitors identified. The identification of a 2′-F-2′-C-methyl cytidine nucleoside PSI-6130 as a safe and selective inhibitor of HCV replication was performed in a replicon-based assay, in which several modifications at the 2′-α and 2′-β nucleoside positions were evaluated, generating valuable structure–activity relationship information. It is worth mentioning that when tested against the BVDV, PSI-6130 did not show any activity [56]. Subgenomic and full-length HCV replicons, as well as the mutant replicon NS5B S282T, have contributed to characterizing the activity of PSI-6130, the lead molecule that originated Sofosbuvir [40,56]. The process of discovery and development of Sofosbuvir, considered a keystone in DAA therapy, is thoroughly described by [57].

Replicon systems have also been a valuable tool to identify anti-HCV compounds targeting proteins considered not druggable due the unavailability of in vitro enzymatic assays, such as NS5A. Despite its essential activity for RNA synthesis, viral assembly, and interaction with host proteins [58,59,60], the NS5A mechanism of action is barely understood and no enzymatic activity has been attributed to this protein. Performing an antiviral screening with Huh7 cells harboring HCV/BVDV dual replicon [20], Bristol-Myers Squibb researchers discovered a class of thiazolidinone derivatives with anti-HCV activity in the low nanomolar range [61]. Although this class of compounds inhibited the HCV RNA replication, it did not inhibit known enzymatic functions of nonstructural proteins, and, therefore, further investigation was carried out. For this purpose, resistant replicon cells were produced and isolated, and the drug-resistance mutations were mapped to the NS5A N-terminal domain. A single mutation with the replacement of NS5A tyrosine at position 93 by a histidine residue (T93H) or a combined mutation with substitution of NS5A leucine to valine at residue 31 (L31V) and glutamine to leucine at position 54 (Q54L) conferred resistance to the investigated thiazolidinone derivatives [61]. In addition, genotype Ia/Ib chimeric replicons [62] were also evaluated using specific NS5A inhibitors, helping to establish the NS5A hyperphosphorylation region as the N-terminal portion involved in inhibitor sensitivity. Further development of the lead compound brought up Daclatasvir, a palindromic compound showing an EC_50_ against Ia/Ib genotype replicons in the low picomolar range [63]. Daclatasvir was approved by the FDA in 2015 to treat chronic hepatitis in a combination therapy with Sofosbuvir, and its discovery and development process took huge advantage of replicon-based assays. Moreover, replicon systems contributed not only to the Daclatasvir-like HCV inhibitor development, but also helped to establish NS5A inhibition as a valuable strategy to block HCV replication. Currently, chronic hepatitis treatment with DAA combination therapy includes one of six NS5A inhibitors: Daclatasvir, Velpatasvir, Ledipasvir, Ombitasvir, Pibrentasvir, and Elbasvir.

### 2.2. Togaviridae Family

The *Togaviridae* family is divided into only two distinct genera: *Alphavirus* and *Rubivirus*. The genus *Alphavirus* comprises viruses transmitted by mosquito vectors, while the genus *Rubivirus* has a unique member, the Rubella virus (RUBV), which is spread by the aerosol route. Alphaviruses are small (60–70 nm in diameter) icosahedral-shaped viruses. The viral genome of about 11–12 kb contains two ORFs encoding two polyproteins: The nonstructural (nsP) or replicase polyprotein, nsP1234, which is processed into nsP1-4; and the structural polyprotein expressed via a subgenomic RNA that generates the structural proteins capsid (C), envelope (E1, E2, E3), and 6K (Figure 2b). The nsPs act in the viral replication complex, and the structural proteins are responsible for viral assembly [64].

#### *Alphavirus* 

Alphavirus infections usually cause fever, rash, and arthritis, which can persist for several months. Since the re-emergence of Chikungunya virus (CHIKV) in 2013–2014, causing more than one million suspected infections in the Caribbean, as well as in the Americas, there has been an increasing interest in identifying inhibitory targets in the CHIKV replication cycle [65].

A BHK-21 cell line harboring a CHIKV replicon expressing enhanced green fluorescent protein (EGFP), Rluc, and Pac was developed and used for the screening of viral replication inhibitors [24]. The Rluc sequence was fused with CHIKV nsP3, and EGFP was produced as a fusion protein with Pac under the control of sg-promoter [24]. This system was validated for antiviral assays in 96 well format and used to test a total of 356 compounds, including 123 natural molecules, 233 clinically approved drugs, and other pharmaceutic compounds. Natural compounds with a 5,7-dihydroxyflavone structure (apigenin, chrysin, naringenin, and silybin) were found to inhibit CHIKV replicon with IC_50_ values ranging from 22.5 to 71.1 µM [24]. Using the same CHIKV replicon cell line, an automated assay was developed in a 384 well microplate format to evaluate about 3000 bioactive compounds, including drugs approved for use as well as those in clinical trials. Of the total, six candidate molecules inhibited the RNA replication. Abamectin, ivermectin, and berberine showed about 85% decrease of the Rluc signal, while cerivastatin, fenretinide, and ivermectin compounds were less efficient, decreasing the luciferase activity in 40% to 70% [65].

Two other studies used the BHK-21 CHIKV-Rluc cell line to evaluate the anti-CHIKV activity of different flavonoids [66,67]. Among quercetin, kaempferol, and silymarin, only the latter was identified with a significant antiviral activity, suppressing the Rluc activity in 93.4% [66]. IC_50_ values of 12, 154.66, and 42.52 µM were obtained for baicalein, fisetin, and quercetagetin, respectively, with fisetin displaying a high cytotoxicity in BHK-21 cells [67]. A Huh7 cell line stably expressing the CHIKV-Rluc replicon [24] was used to evaluate the effect of two class II cationic amphiphilic drugs in the viral replication [68]. Imipramine, an FDA-approved antidepressant drug, showed no cytotoxic effects and displayed a dose-dependent decrease in Rluc activity [68]. Another BHK-21 cell line harboring a Chikungunya virus replicon, derived from the infectious clone 5-pCHIKic, expressing EGFP and Fluc (CHIKrep-FlucEGFP) was developed by [69] and later used to evaluate the antiviral effects of suramin, an approved anti-parasitic drug [70]. The compound inhibited RNA synthesis and expression of the EGFP [70].

Two types of Nano luciferase (NanoLuc) CHIKV replicon, a wild type and an RdRp-inactivated replicon, were constructed and used to transfect HEK-293T (human embryonic kidney) cells to test the viral replication efficiency in the presence of Compound-A, a molecule containing a benzimidazole structure that was able to inhibit CHIKV infection at nanomolar concentrations [71]. The NanoLuc activities were measured in cell lysates at 6, 12, 24, and 36 h post-transfection (h.p.t.), and viral genome replication of the wild-type replicon was found to be significantly inhibited at 24 and 36 h.p.t. by 106.0 and 216.7 times compared to that in the absence of Compound-A [71]. The NanoLuc activity of the RdRp-inactivated replicon was stable from 6 to 36 h.p.t., therefore demonstrating that the RdRp function was precisely inactivated and the RNA was translated until 6 h.p.t. In addition, genome translation efficiency was not affected by Compound-A. Thereby, the results demonstrated that Compound-A significantly inhibited CHIKV genome replication by targeting nsP4 protein, but did not inhibit genome translation [71].

### 2.3. Coronaviridae Family

Members of *Coronaviridae* family are the largest known RNA viruses, with 118–140 nm in diameter and with typical prominent spikes in their viral envelope. The genome of ~25–32 kb contains two major ORFs, 1a and 1b, which encode 16 nonstructural proteins (nsp1 to 16). ORF1b is transcribed after a −1 ribosomal frameshift. The structural proteins (S, spike; E, envelope; M, membrane; N, nucleocapsid) and accessory proteins are expressed from subgenomic RNAs [72] (Figure 2c).

The family is divided into two subfamilies, the *Coronavirinae* and the *Torovirinae*. In the *Coronavirinae* subfamily, the genus *Coronavirus* includes SARS-CoV and MERS-CoV, both responsible for causing serious respiratory diseases in humans with high morbidity and mortality rates [73,74]. The ongoing COVID-19 (coronavirus disease 2019) outbreak, caused by a novel coronavirus SARS-CoV-2, emerged from Wuhan, Hubei province of China, and further spread to at least 115 countries, being declared by the WHO as a “Public Health Emergency of International Concern” on January 30, 2020 [75] and as a global pandemic on March 11, 2020 [76].

#### *Coronavirus* 

A SARS-CoV (Urbani strain) DNA replicon was developed as a BAC system under the control of the CMV promoter due to the large size of the genome and the instability of some CoV replicase gene sequences in bacteria. The replicon system was constructed with the 5′ and 3′ ends of the viral genome, the replicase, and the N genes [21]. HEK-293T cells were transfected with replicon DNA and used to confirm the effect of the triazole derivative SSYA10-001, a noncompetitive inhibitor of viral helicase (nsp13), in decreasing viral replication. SSYA10-001 showed an inhibitory activity (EC_50_ value of 8.95 μM) with low cytotoxicity (CC_50_ > 250 μM), suggesting that the helicase plays a still-unidentified critical role in the SARS-CoV life cycle [77]. The compound was also shown to exhibit a broad-spectrum activity against other coronaviruses, such as MERS-CoV and Mouse Hepatitis Virus (MHV) [78].

An HTS assay using a BHK-21 SARS-CoV replicon cell line was performed to screen a total of 7035 small-molecule compounds. SARS-CoV RNA replicon construct was developed from the SIN2774 strain genome by the deletion of the envelope coding genes S, E, and M, and insertion of a GFP co-expressed from a polyprotein containing the blasticidin-resistant gene (BlaR) selectable marker [25]. Seven compounds were identified to display an anti-SARS-CoV activity, with IC_50_ values ranging from 1.4 to 5.8 µM [79]. Another RNA replicon system in which the S gene was replaced by the EGFP was used to transiently transfect HEK-293T cells. Inhibition assays using ribavirin and the cysteine proteinase inhibitor E64-D at 12 h.p.t. showed that the E64-D was able to decrease EGFP expression at 60 h after treatment, indicating that the system can be used as a tool for drug screening [73].

## 3. Non-Enveloped Viruses

### 3.1. Picornaviridae Family

The *Picornaviridae* is one of the largest viral families, classified in eight established and five proposed genera [80]. Picornaviruses are non-enveloped icosahedral viruses responsible for causing a variety of diseases going from mild respiratory illness (rhinoviruses) to acute flaccid paralysis—AFP (polioviruses) [81]. The viral genome of about 7.0 to 8.5 kb is composed of a single ORF flanked by 5′ and 3′ UTR’s and is covalently linked to the VPg (3B) at its 5′ terminus as well as polyadenylated at its 3′ terminus [82] (Figure 3a). The ORF encodes a single polyprotein, which, following synthesis, undergoes multiple cleavages by the two viral proteases 2A^pro^ and 3C^pro^ to produce the final mature viral capsid proteins and nonstructural proteins [81].

The best-known picornaviruses are EV-71, PV, Hepatitis A virus (HAV), and Human rhinovirus (HRV) [80]. Currently, only PV and HAV can be controlled by vaccination, and, therefore, antiviral therapies would be highly desirable for many picornavirus-associated diseases for which there are no approved antiviral drugs [86].

#### 3.1.1. *Enterovirus*

Enteroviruses are ubiquitous and are common human pathogens, classified into four species formerly denominated as human enteroviruses A to D (HEV-A to HEV-D) [87]. The HEV-A members Coxsackievirus A16 (CVA16) and EV-71 are the main causative agents of hand, foot, and mouth disease (HFMD). EV-71 infections have often been associated with severe outcomes involving neurological or cardiopulmonary complications and even death [88,89]. Among HEV-B enteroviruses, coxsackieviruses type B (CV-B), and mainly Coxsackievirus B3 (CV-B3) are considered to be a common cause of acute myocarditis in children and young adults [90,91]. Poliovirus (PV), the best-studied member of the HEV-C species, is the causative agent of paralytic poliomyelitis. Although polio has been officially eradicated in Europe and in the Americas, PV is still endemic in several countries and regions [87].

Several compounds with anti-enterovirus activity have been evaluated in clinical trials, including Rupintrivir, a viral 3C protease inhibitor, and the small-molecule capsid-binding inhibitors Pirodavir and Pleconaril. However, these compounds are no longer being developed as antiviral drugs due to their limited efficacy in the context of natural infection, safety concerns, or potential drug interactions [92].

Many enterovirus subgenomic RNA replicon constructs harboring a Fluc gene in place of the capsid-code P1 region were developed for antiviral assays [87,93,94,95]. The antiviral effect of the thiazolobenzimidazole TBZE-029 in CV-B3 replication was evaluated in a transient assay using Buffalo green monkey (BGM) cells transfected with pCB3/T7-Luc replicon RNA in the presence (at 80 µM concentration) or absence of the compound [93]. TBZE-029 almost completely inhibited the accumulation of viral positive-strand RNA, as evidenced by a 50-fold decrease in the Fluc activity. In this study, TBZE-029 was characterized as an inhibitor of the 2C protein, although the precise mechanism of action was not described [93]. Another CV-B3 inhibitor compound, the quinoline derivative Golgicide A (GCA), which specifically inhibits Golgi-specific BFA resistance factor 1 (GBF1), was tested using pCB3/T7-Luc replicon [94]. BMG cells were transfected with replicon RNA and subsequently treated with GCA. At 2, 4, 6, or 8 h.p.t. GCA was found to strongly inhibit Fluc accumulation, demonstrating that this compound hinders viral RNA replication [94].

A primary screening of 400 highly purified natural compounds was performed by Xu and colleagues (2014) using both subgenomic replicons and reporter EV-71 and CVA16 viruses [95]. A total of 44 compounds were identified to strongly reduce EGFP and/or luciferase expression (fluorescence and/or luminescence value < 0.5) and were further evaluated by cell-viability-based secondary screening. Using CPE, plaque assay, and reporter virus-based assay, similar EC_50_ values were obtained for the flavonoid derivative luteolin, the most promising hit, with SI values of 14.36 and 20.03 for EV-71 and CVA16, respectively, suggesting that the molecule is a potential candidate for antiviral therapy [95].

EV-71 replicon was used in a transient assay to validate the antiviral activity of itraconazole (ITZ), a triazole antifungal agent [87]. BHK-21 cells were transfected with replicon RNA, seeded in a 12 well plate, and immediately treated with 10 µM ITZ. This compound was found to decrease the Fluc activity at 16 h.p.t. at 89.2%, demonstrating that ITZ strongly suppresses viral RNA replication or polyprotein processing [87]. Another study evaluated a total of 968 FDA-approved drugs as potential antivirals using both EV-71 and CV-B3 replicon systems [96]. Vero cells were transfected with EV-71 replicon RNA and simultaneously treated with the compounds at 10 µM concentration. Micafungin, a well-known antifungal drug, showed an IC_50_ value of ~5–8 µM and displayed a low cytotoxicity. Similar results were obtained for Vero cells transfected with CV-B3 replicon RNA. Nevertheless, the precise mechanism of the antiviral effect of Micafungin on EV-71 was not determined [96].

The antiviral activity of compounds from a natural product library (BML-2865, Enzo Life Sciences) was evaluated in a replicon-based transient assay using EV-71 NanoLuc replicons [89]. Camptothecin, a DNA topoisomerase 1 (TOP1) inhibitor, was found to significantly inhibit the accumulation of NanoLuc in cells expressing both replication-competent and replication-incompetent replicons. The replication-incompetent construct has a 53 amino acid deletion from the C-terminus of viral RdRp 3D (3D^del^) that prevents RNA replication. Therefore, the results suggested an inhibitory mechanism not specific to viral RNA replication alone, but likely implicate viral RNA translation as well [89].

A poliovirus RNA replicon with the Rluc sequence placed in frame with the polyprotein was developed by [83], and was later used for antiviral screenings. A transient replication assay was performed to evaluate the anti-PV activity of 192 cell-permeable kinase inhibitors from Calbiochem. HeLa cells were transfected with replicon RNA and replication was monitored in live cells incubated with 10 µM of the compounds for 16 h [86]. Three compounds, Akt inhibitor IV (A4(1)), PDGF receptor tyrosine kinase inhibitor III (E5(1)), and indirubin derivative E804 E7(2), showed IC_50_ values of 3.2, 12, and 16 µM, respectively. However, a significant cytotoxicity was observed for A4(1) at 50 µM concentration. All three compounds were effective against PV, the related CV-B3, as well as the much more distantly related EMCV from the *Cardiovirus* genus, evidencing the broad-spectrum effect of those molecules. Evaluation of A4(1) in vivo in a murine model of poliomyelitis showed that, although this compound had significant toxicity, it demonstrated a protective anti-poliovirus potential by delaying the development of the disease [86].

To date, more than 160 HRV genotypes have been isolated and classified into the major groups of HRV-A, -B, and -C based on phylogenetic relationships [97,98]. HRV infections cause a broad spectrum of respiratory illnesses in humans, varying from a common cold to a severe and fatal pneumonia in susceptible populations [99]. Unlike HRV-A and -B, which can be propagated in conventional cell lines, HRV-C viruses were shown to only replicate in specific human airway tissues, hampering the development of screening assays for anti-HRV drugs [99,100].

A cell-based assay that can be used for antiviral screenings was successfully developed in 2014 by Mello and colleagues using subgenomic replicon systems for HRV-C. Antiviral activities of the capsid inhibitor Pleconaril, phosphatidylinositol 4-kinase III (PI4K-III) inhibitor PIK93, nucleoside analog inhibitor of 3D polymerase MK-0608, and the 3C protease inhibitor Rupintrivir were evaluated in a transient assay using H1-HeLa cells transfected with replicon constructs for genotypes HRVc24, HRVc25, HRVc15, and HRVc11 containing an Rluc sequence in the P1 capsid region. With the exception of Pleconaril, compounds inhibited the viral replication in a dose-dependent manner at low nanomolar ranges. To establish a high-throughput HRV-C antiviral screening, the HRVc15 replicon assay was optimized to a 384 well format. The results showed a robust and reproducible luciferase assay, supporting the use of this system to identify novel pan-serotype human rhinovirus inhibitors [99].

#### 3.1.2. *Hepatovirus*

The Hepatitis A virus is the sole member of the *Hepatovirus* genus. HAV causes acute hepatitis and occasionally fulminant hepatitis, a life-threatening disease with a significant morbidity worldwide [101]. The risk of infection exists in countries lacking HAV immunity or where the endemicity of hepatitis A is low or intermediate, causing outbreaks that could be difficult to control. Although HAV vaccination is efficient, new therapeutic options are desired to control HAV infections [101].

The ability of amantadine, a tricyclic symmetric amine, to inhibit HAV IRES-mediated translation in Huh7 cells was reported by Kanda and colleagues in 2005, and was then evaluated in a replicon-based assay by the same group in 2010 [101,102]. Huh7 cells were transfected with pSV40-HAV-IRES vector encoding SV40 promoter-driven Rluc and Fluc, separated by HAV-IRES [102], and treated with amantadine and/or IFN-α. Inhibition of luciferase activity was observed with amantadine with or without 100 IU/mL IFN-α. To further characterize those inhibitory effects, a DNA replicon that stably expresses T7 RNA polymerase or a replication-incompetent HAV replicon were transfected into HuhT7 cells, and the drugs were added 24 h later. The results showed that the combination of amantadine and IFN-α can suppress HAV replication more efficiently than amantadine or IFN-α alone, suggesting a potential use of this drug combination to treat acute hepatitis A [101].

### 3.2. Hepeviridae Family

The *Hepeviridae* family is divided into two genera: *Orthohepevirus*, with four separate species (*Orthohepevirus A–D*), including viruses in mammals and birds, and *Piscihepevirus*, with a single virus species in fish [103]. Virions are icosahedral, non-enveloped, and spherical particles with a diameter of 27 to 34 nm [103]. The genome of about 7.5 kb contains three ORFs: ORF 1, which encodes nonstructural proteins required for viral replication; ORF 2, which encodes for capsid protein; and, ORF 3, which overlaps with ORF 2 and encodes for a phosphoprotein required for viral infectivity [104] (Figure 3b).

The Hepatitis E virus from the *Orthohepevirus* genus is the causative agent of hepatitis E, a self-limiting, acute, and rarely fatal disease in young adults [103]. HEV infection can be severe in patients with liver diseases and has been associated with high mortality rates among children under two years of age [105]. In addition, the mortality rates in pregnant woman reach 30%, and infection can lead to chronicity in immunocompromised patients [106]. The only drug available for hepatitis E is ribavirin; still, it cannot be used by pregnant women due to the risk of fulminant hepatic failure and spontaneous abortion. Therefore, there is a need for the development of safe, non-teratogenic, and effective treatment against HEV infection [106].

Recently, two different HEV replicon systems were developed for antiviral assays. An HEV human genotype 1 (G1) Fluc replicon (pSK-HEV-2-Luc) was used to test 16 antivirals that target the RdRp [84]. NITD008 and Sofosbuvir, both nucleoside analogs, as well as GPC-N114, a non-nucleoside inhibitor, decreased the luciferase signal in a dose-dependent manner and were not toxic to the Huh7 cells, as demonstrated by the achieved SI values of >3000, >51, and >93, respectively. Next, NITD008 and GPC-N114 were evaluated in combination against HEV replicon, and demonstrated a potent antiviral activity and combinational synergy in vitro (mean combination index of 0.4), showing that both compounds could provide useful scaffolds for further antiviral development against HEV [84]. Another replicon construct, pJE03-1760F/P10-Gluc, was used to screen 767 FDA-approved compounds. In this system, the ORF 2 and ORF 3 genes were replaced by Gluc, and PLC/PRF/5 cells were used in the assay [107]. As a result, the inhibitory effects of 20 compounds were confirmed; among them, the antibiotic ciprofloxacin inhibited HEV replication and displayed low cytotoxicity. In addition, IFN-λ1–3 were tested and also inhibited viral replication [107]. Although ciprofloxacin exhibited an inhibitory effect in the reporter system, the drug did not sufficiently inhibit HEV growth in the in vitro culture. [107] Nevertheless, the improvement of new replicon systems for HEV could help in the search for novel compounds to treat hepatitis E.

### 3.3. Caliciviridae Family

The *Caliciviridae* family comprises small icosahedral viruses with a capsid displaying cup-shaped depressions and diameters between 35 and 39 nm. The genome of ~7.3 to 9.0 kb, with a VPg covalently linked to the 5′ terminus and polyadenylated at its 3′ terminus, contains two or three ORFs (Figure 3c). The nonstructural proteins are encoded by genome-length mRNA, and the capsid protein by subgenomic mRNA [108]. The family is divided into five genera, with two of them, *Sapovirus* genus and *Norovirus* genus, comprising human pathogens that cause acute gastrointestinal diseases [108].

Norwalk virus (NV), from the *Norovirus* genus, is the number one cause of foodborne illnesses around the world, responsible for extensive outbreaks of gastroenteritis [109]. The infection is generally acute and self-limiting; however, severe dehydrating diarrhea that could be fatal can occur in children, elderly, and immunocompromised individuals [110]. Regarding the treatment, there are no specific antivirals or vaccines available to prevent NV infection [110].

The major limitation for the discovery of specific compounds against noroviruses is the lack of an efficient and suitable cell culture model [110]. To develop a cell-based assay for antiviral screenings, a human norovirus replicon cell line was established by Chang et al. (2006) by transfecting a Norwalk virus RNA replicon, pNV-Neo, in which the ORF 2 was replaced by the Neo gene in Huh7 cells [85]. Viable Huh7 cell colonies bearing the NV replicon were selected with G418, and a cell clone designated HG23 was selected for further study. The replicon cell line was then incubated with increasing concentrations of IFN-α, and the results showed that the treatment inhibited protein expression in a dose-dependent manner and no cytotoxic effects were observed [85]. The HG23 cell line was also used by the same group to test the effect of IFN-γ and ribavirin on viral replication [111]. IFN-γ was found to inhibit RNA replication in a dose-dependent manner, as previously reported for IFN-α. Ribavirin also decreased viral replication, and the combination with IFN-γ showed an additive effect on the inhibition of viral replication [111]. Thus, the results demonstrated that IFNs and ribavirin could be therapeutic options for the treatment of noroviral gastroenteritis, and confirm that the NV replicon system could be used for the screening of antiviral compounds [111].

In an effort to discover antiviral drugs against norovirus infection, different compound series were tested using the HG23 cell line. After an initial screening of peptide-conjugated phosphonodiamidite morpholino oligomers (PPMOs) using a murine norovirus (MNV) cell system, *Noro 1.1*, a PPMO that specifically inhibited MNV replication, also decreased protein expression and, consequently, the replication of NV replicon [112]. Another five compounds from a series of 22 amino-acid-derived acyclic sulfamide-based molecules were found to display potent inhibitory activities and low cytotoxicity against NV replicon [113]. A total of eight dipeptidyl α-hydroxyphosphonates compounds were tested in the NV replicon cell line, and two of them showed a high inhibitory potency [114]. A compound series of tripeptidyl transition state inhibitors were synthesized and assessed for inhibitory effects using the Norwalk virus replicon cell line at concentrations ranging from 0.01 to 10 μM [115]. Additionally, the compounds were tested directly against norovirus protease using a FRET-based kinetic assay to test for enzymatic activity. As results, four peptidomimetic compounds displayed anti-noroviral activity, decreasing replication in the sub-micromolar range [115]. Although those compounds are potential antivirals, toxicity assays must be performed to confirm their safety.

More recently, Harmalkar and colleagues (2019) reported the identification of non-nucleoside vinyl-stilbene analogs as potent norovirus replication inhibitors [116]. Some compounds were subjected to structural modifications via an empirical medicinal chemistry approach to improve their anti-noroviral activity. Then, new replication inhibition assays were performed using the HG23 cell line, and the results showed that vinyl stilbene compounds may be optimal for inhibitory activity against norovirus replication [116]. The use of the NV replicon system in the studies presented herein enabled the identification of different classes of compounds with antiviral activities and may facilitate the development of effective anti-norovirus therapeutics in the near future. With regard to the *Sapovirus* genus, the development of replicon systems has not been reported for any member yet.

## 4. Conclusions

Over the past decades, several replicon-based systems have been developed for the discovery of DAA targeting (+) ssRNA viruses (Figure 4). Despite the genetic modifications inserted into the viral genome sequence, including the deletion of partial or entire structural protein coding sequences and insertion of different reporter genes, the system remains a safe and reliable tool for screening large compound libraries. This may be exemplified by HCV, for which the use of reporter replicon systems was pivotal for the discovery of new DAAs, such as Sofosbuvir and Daclatasvir, both approved by the FDA to treat chronic hepatitis in a combination therapy [57,63]. Moreover, replicon-based assays have also been a valuable tool to identify antiviral compounds targeting proteins considered not druggable due to the lack of in vitro enzymatic assays, like HCV NS5A [63]. Notwithstanding, development of cell-based assays for some viruses that proved difficult to culture in common cell lines, such as HRV-C viruses and noroviruses, was made possible through the establishment of replicon systems, allowing the screening for inhibitors of the viral replication. Finally, the use of reporter replicons has enabled the identification of several small-molecule compounds displaying effective antiviral activities on low nanomolar ranges with high selectivity, providing useful scaffolds for further DAA development against enveloped and non-enveloped (+) ssRNA viruses (Table 1).

## Figures and Tables

**Figure 1 viruses-12-00598-f001:**
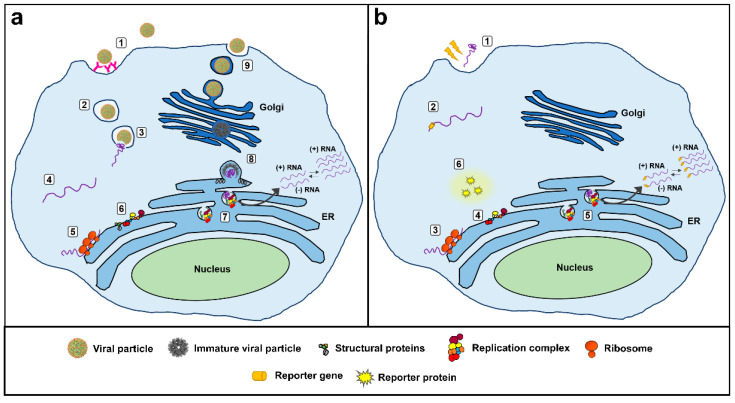
Schematic representation of a viral infection (**a**) compared to the transfection of cells with subgenomic RNA replicon systems (**b**). In (**a**), viral particles (1) are recognized by cell receptors and enter the cell via endocytosis (2). The acid environment of endosomes leads to the reorganization of the envelope glycoprotein (3), allowing the fusion of viral and endosomal membranes and the release of viral (+) ssRNA into the cytosol (4). Viral RNA hijacks the host cell’s translation machinery (5) to produce structural and nonstructural proteins (6). The nonstructural proteins assemble at the endoplasmic reticulum to form the replication complex together with host cell factors (7), allowing the viral RNA replication. Nascent viral RNA is englobed by viral structural proteins, forming immature viral particles (8). Immature viral particles travel through the Golgi apparatus, where pH and enzymatic modifications allow particle maturation and release. In (**b**) in vitro transcribed replicon RNA (1) is transfected into susceptible mammalian cells. In this case, electroporation is represented by the rays because it is the most widely used physical method able to transfect a large number of cells in a short time, as it is an easy and rapid transfection option [16]. (2) Replicon RNA harbors a reporter gene in place with structural protein genes. (3) Replicon RNA hijacks the host cell’s translation machinery to produce the reporter protein and nonstructural proteins. (4) Nonstructural proteins assemble at the endoplasmic reticulum to form the replication complex together with host cell factors, allowing the replicon RNA replication (5). (6) The reporter protein in the cytoplasm produces a fluorescent or luminescent signal that correlates with the level of accumulated replicon RNA, allowing the follow-up of the replicon replication efficiency.

**Figure 2 viruses-12-00598-f002:**
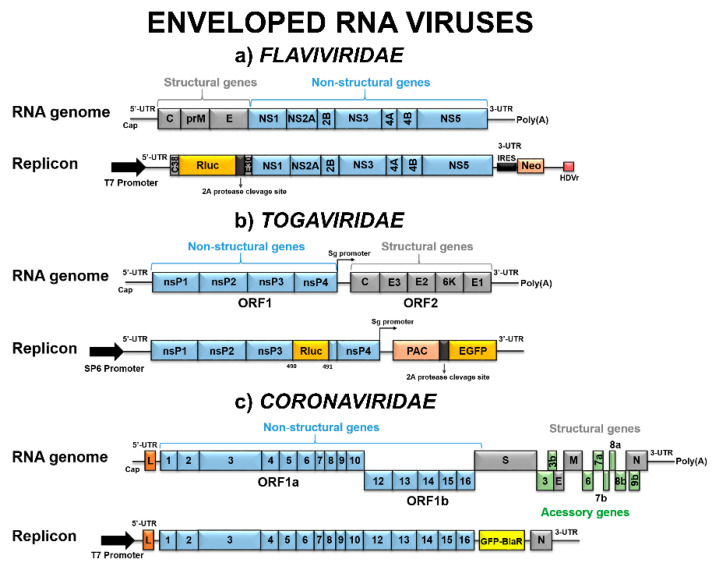
Schematic representation of enveloped RNA viruses’ genome and subgenomic replicon systems. (**a**) *Flaviviridae* replicon based on the construct described by [8] for the Zika virus (ZIKV). C_38_ and E_30_ represent DNA sequences encoding the first 38 amino acids of C protein and the last 30 amino acids of E protein, respectively. A fragment containing the internal ribosomal entry site (IRES) and Neo gene was inserted downstream of the first 28 nucleotides of 3′UTR. Rluc—*Renilla* luciferase; HDVr—hepatitis delta virus ribozyme sequence. (**b**) *Togaviridae* replicon based on the construct described by [24] for the Chikungunya virus (CHIKV). A cassette encoding Pac fused to the enhanced green fluorescent protein (EGFP) via the 2A autoprotease of the foot-and-mouth disease virus (FMDV) was inserted under the control of the sg-promoter. In addition, the coding sequence of Rluc was inserted after the codon for amino acid 1823 of P1234 reading frame (after codon 490 of nsP3). (**c**) *Coronaviridae* replicon based on the construct described by [25] for Severe Acute Respiratory Syndrome Coronavirus (SARS-CoV). The nucleocapsid gene, *N*, was retained because the encoded protein has been shown to be required for viral RNA synthesis. Green fluorescent protein–blasticidin deaminase fusion (*GFP–BlaR*) gene was inserted between ORF 1 and *N*, not at the 5′ or 3′ end of the replicon, in order to minimize any possible deleterious effects in the synthesis of replicon RNA. L—leader sequence.

**Figure 3 viruses-12-00598-f003:**
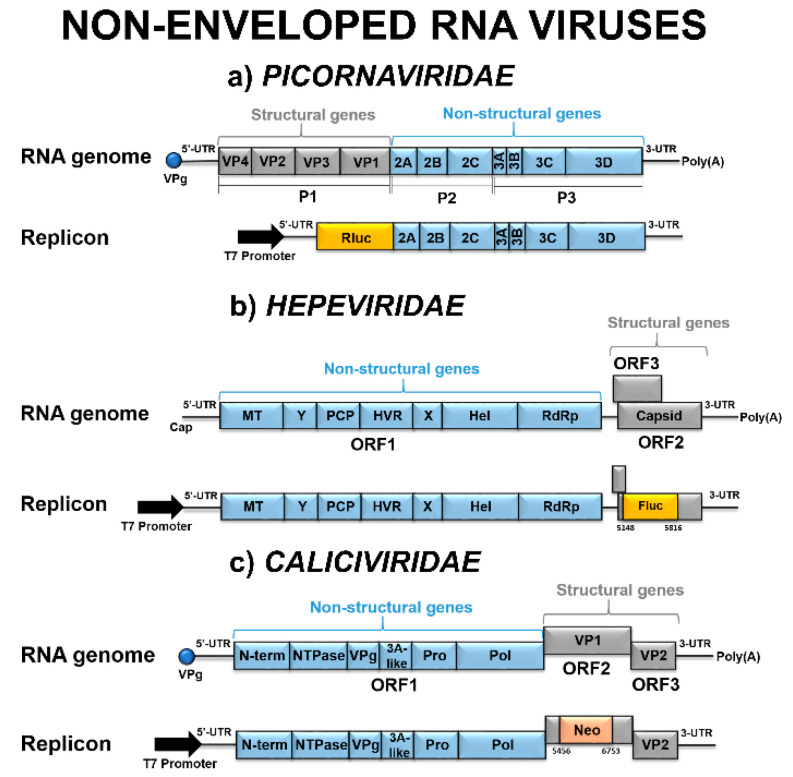
Schematic representation of non-enveloped RNA viruses’ genomes and subgenomic replicon systems. (**a**) *Picornaviridae* replicon based on the construct described by [83] for the Poliovirus (PV). The capsid coding P1 region is replaced by the Rluc gene. (**b**) *Hepeviridae* replicon based on the construct described by [84] for the Hepatitis E virus (HEV). The ORF2 capsid gene is disrupted with the Fluc gene (nucleotides 5148 to 5816). (**c**) *Caliciviridae* replicon based on the construct described by [85] for the Norwalk virus (NV). The Neo gene was engineered into the 5′-end region of the ORF2 so that the expressed product would contain the first 33 aa of the NV VP1 fused in frame with neomycin phosphotransferase.

**Figure 4 viruses-12-00598-f004:**
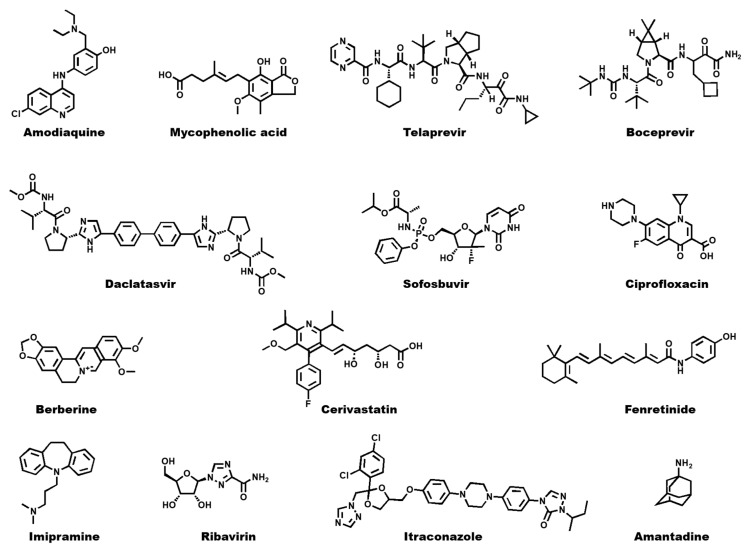
Two-dimensional structures of approved antivirals against single-stranded positive RNA ((+) ssRNA) viruses.

**Table 1 viruses-12-00598-t001:** Compounds identified as antiviral inhibitors using replicon-based assays.

Virus	Inhibitor	Current Status
**DENV**	Triaryl pyrazoline [26]	In vitro assays
	Amodiaquine (AQ) [33]	Approved
	Compound C and tolcapone [28]	In vitro assays
	Tannic acid [29]	In vitro assays
**WNV**	Compound CDDMN [10]	In vitro assays
	Compound B [28]	In vitro assays
**YFV**	CCG-4088 and CCG-3394 [31]	In vitro assays
**ZIKV**	Mycophenolic acid (MPA) [9]	Approved
	2′-C-methyladenosine (2′-C-MeAdo) [9]	In vitro assays
	6-azauridine (6-AzUrd) [9]	In vitro assays
**HCV**	BILN 2061 [54]	Pre-clinical trials (discontinued)
	Telaprevir [55]	Approved
	Boceprevir [55]	Approved
	PSI-6130 (lead molecule of Sofosbuvir) [40,56]	Approved (Sofosbuvir)
	Thiazolidinone derivatives [61]	In vitro assays
	Daclatasvir [63]	Approved
**CHIKV**	5,7-dihydroxyflavone derivatives [64]	In vitro assays
	Abamectin [65]	Approved
	Ivermectin [65]	Approved
	Berberine [65]	Approved
	Cerivastatin [65]	Approved
	Fenretinide [65]	Approved
	Silymarin [66]	Approved
	Imipramine [68]	Approved
	Suramin [70]	Approved
	Compound-A [71]	In vitro assays
**SARS-COV**	Compound SSYA10-001 [77]	In vitro assays
	Ribavirin [73]	Approved
	E64-d [73]	In vitro assays
**CV-B3**	Thiazolobenzmidazole (TBZE-029) [93]	In vitro assays
	Golgicide A (GCA) [94]	In vitro assays
	Micafungin [96]	Approved
**EV-71**	Luteolin [95]	In vitro assays
	Itraconazole [87]	Approved
	Micafungin [96]	Approved
	Camptothecin [89]	In vitro assays
**CVA16**	Luteolin [95]	In Vitro assays
**PV**	Akt inhibitor IV (A4(1)) [86]	Pre-clinical trials
	PDGF receptor tyrosine kinase inhibitor III (E5(1)) [86]	In vitro assays
	Indirubin derivative E804 E7(2) [86]	In vitro assays
**HRV-C**	Pleconaril [99]	Clinical trial (discontinued)
	PIK93 [99]	In vitro assays
	MK-0608 [99]	In vitro assays
	Rupintrivir [99]	Clinical trials
**HAV**	Amantadine [101]	Approved
	IFN-alpha [101]	Approved
**HEV**	NITD008 [84]	In vitro assays
	Sofosbuvir [84]	Approved
	GPC-N114 [84]	In vitro assays
	Ciprofloxacin [107]	Approved
	IFN-gama 1–3 [107]	Approved
**NV**	IFN-alpha [85]	Approved
	IFN-gama [111]	Approved
	Ribavirina [111]	Approved
	Peptide-conjugated phosphonodiamidite morpholino oligomers (PPMOs) [112]	In vitro assays
	Amino acid-derived acyclic sulfamide-based compounds [113]	In vitro assays
	Dipeptidyl α-hydroxyphosphonates compounds [114]	In vitro assays
	Tripeptidyl transition state compounds [115]	In vitro assays
	Non-nucleoside vinyl-stilbene analogs [116]	In vitro assays

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
