# Peer review of "Reporter Replicons for Antiviral Drug Discovery against Positive Single-Stranded RNA Viruses"

_viruses, 2020, doi:10.3390/v12060598_

Round 1

Reviewer 1 Report

The manuscript under review, “Reporter replicons for antiviral drug discovery against positive single-stranded RNA viruses,” discusses utilization of replicons for screening for antiviral agents against a number of families of RNA viruses. As such, it can serve as a nice introduction to the historical development and use of these reagents for drug discovery. However, the manuscript could be significantly improved by including additional discussion of the advantages and disadvantages of various approaches to replicon development as well as a bit more information regarding minimal requirements for useful constructs and, finally, discussion of complementary approaches.

Minimal requirements: The figures in the manuscript nicely illustrate the general approach. However, it is not clear either from the figures or from the text if there are minimal requirements for constructing active replicons (e.g., which components are necessary for a construct to “work”?). Similarly, are these minimal requirements consistent across the families of viruses discussed? What are the functional consequences of adding or removing specific genetic elements?

Alternatives for replicon development: Several alternative approaches to constructing replicons are mentioned in the examples discussed in the text and in many of the references. It would be useful to have a more comprehensive general discussion of (for example) advantages and disadvantages of delivery of RNA (and options used – only electroporation appears in the figure) as compared to cDNA. Similarly, comparisons of cells transiently transfected with cDNA vs selection of stable cell lines would be useful.

Alternative detection approaches: Multiple reporter constructs for detection/quantitation of replication are mentioned. It would be of value to have a more detailed discussion, again assessing both the various alternatives and their advantages and disadvantages.

Potential pitfalls: As with any HTS approach, use of replicons can (and will) lead to false positive and false negative results. These should be discussed in general terms and commonly used secondary assays for confirmation and mechanistic studies should be briefly mentioned.

Complementary approaches to studying RNA viruses: The most obvious of these would be use of pseudovirus particles to investigate viral particle binding and initial infectivity, but there are others in the literature as well. Complementary approaches have the potential for generating combination therapies for target viruses and should be mentioned.

Summary table: I counted at least 14 specific viruses mentioned in the text as having been investigated using the replicon approach (I may have missed some). A summary table listing all of the viruses thus assessed along with compounds identified by screening with replicons and the status of those compounds would be extremely useful. Alternatively, a table restricted to compounds in clinical or pre-clinical development, while less useful, would be of value. Or, a summary of just the compounds shown at the end of the manuscript would be useful.

In general, these suggested additions could be accommodated by including tables and/or supplementary information with a paragraph or two of discussion. As noted above, the review could be of some value as it currently stands, but additional information would make it considerably more useful as a guide to options available for further investigation. Most of the additional information is already present in the text and/or in references and the examples are good, but need to be organized in a more systematic way.

A somewhat more comprehensive review would be a much more valuable contribution to this timely topic.

Author Response

Point-by-point Response to Reviewer 1 Comments

The manuscript under review, “Reporter replicons for antiviral drug discovery against positive single-stranded RNA viruses,” discusses utilization of replicons for screening for antiviral agents against a number of families of RNA viruses. As such, it can serve as a nice introduction to the historical development and use of these reagents for drug discovery. However, the manuscript could be significantly improved by including additional discussion of the advantages and disadvantages of various approaches to replicon development as well as a bit more information regarding minimal requirements for useful constructs and, finally, discussion of complementary approaches.

Point 1: Minimal requirements: The figures in the manuscript nicely illustrate the general approach. However, it is not clear either from the figures or from the text if there are minimal requirements for constructing active replicons (e.g., which components are necessary for a construct to “work”?). Similarly, are these minimal requirements consistent across the families of viruses discussed? What are the functional consequences of adding or removing specific genetic elements?

Response 1: We thank the reviewer for raising these questions. The minimal requirements to construct replicon systems for (+) ssRNA viruses, are consistent across the families discussed in the manuscript, as described in more details in [1–3]. A brief discussion about those requirements were added to the text (lines 45 to 58). In respect to the consequences of adding or removing specific genetic elements, an explanation was also added to the manuscript (lines 90 to 95).

Point 2: Alternatives for replicon development: Several alternative approaches to constructing replicons are mentioned in the examples discussed in the text and in many of the references. It would be useful to have a more comprehensive general discussion of (for example) advantages and disadvantages of delivery of RNA (and options used – only electroporation appears in the figure) as compared to cDNA. Similarly, comparisons of cells transiently transfected with cDNA vs selection of stable cell lines would be useful.

Response 2: For clarification, we added a brief discussion about the advantages/disadvantages of RNA delivery as compared to cDNA (lines 58-63) and justified the use of electroporation (lines 114-116). The comparison of cells transiently transfected vs. stable cell lines was also added to the text (lines 73-75; and, lines 79-81), as suggested by the reviewer.

Point 3: Alternative detection approaches: Multiple reporter constructs for detection/quantitation of replication are mentioned. It would be of value to have a more detailed discussion, again assessing both the various alternatives and their advantages and disadvantages.

Response 3: The authors thank the reviewer for the suggestion; however, the use of either fluorescent reporter proteins or luciferases are suitable for antiviral HTS, as discussed in the manuscript. Both strategies allow the quantitative measure of the effects of replication inhibitors on the reporter protein expression/activity, that is the main goal of the replicon-based assay. A detailed discussion about fluorescent and luminescent detection methods would be very extensive and we believe that would not be as much informative to the comprehension of the replicon system. The principles of a bioluminescent assay and the advantages/disadvantages of its use for HTS in comparison to fluorescent assays can be found in details in [4].

Point 4: Potential pitfalls: As with any HTS approach, use of replicons can (and will) lead to false positive and false negative results. These should be discussed in general terms and commonly used secondary assays for confirmation and mechanistic studies should be briefly mentioned.

Response 4: As suggested by the reviewer, a brief mention on the potential pitfall of the use of replicon-based assays and secondary confirmation assay were added to the text (lines 75-79).

Point 5: Complementary approaches to studying RNA viruses: The most obvious of these would be use of pseudovirus particles to investigate viral particle binding and initial infectivity, but there are others in the literature as well. Complementary approaches have the potential for generating combination therapies for target viruses and should be mentioned.

Response 5: An alternative approach to the use of replicons was added to the manuscript (lines 83 to 89), as suggested by the reviewer.

Point 6: Summary table: I counted at least 14 specific viruses mentioned in the text as having been investigated using the replicon approach (I may have missed some). A summary table listing all of the viruses thus assessed along with compounds identified by screening with replicons and the status of those compounds would be extremely useful. Alternatively, a table restricted to compounds in clinical or pre-clinical development, while less useful, would be of value. Or, a summary of just the compounds shown at the end of the manuscript would be useful.

Response 6: A summary table (Table 1) listing all the viruses along with the compounds identified by screening with replicons was added to the text (lines 629-630), as suggested by the reviewer.

In general, these suggested additions could be accommodated by including tables and/or supplementary information with a paragraph or two of discussion. As noted above, the review could be of some value as it currently stands, but additional information would make it considerably more useful as a guide to options available for further investigation. Most of the additional information is already present in the text and/or in references and the examples are good, but need to be organized in a more systematic way.

A somewhat more comprehensive review would be a much more valuable contribution to this timely topic.

References:

  1. Tews, B.A.; Meyers, G. Self-Replicating RNA. Methods Mol. Biol. 2017, 1499, 15–35, doi:10.1007/978-1-4939-6481-9_2.
  2. Kümmerer, B.M. Establishment and Application of Flavivirus Replicons. In Advances in experimental medicine and biology; 2018; Vol. 1062, pp. 165–173.
  3. Aubry, F.; Nougairede, A.; de Fabritus, L.; Querat, G.; Gould, E.A.; de Lamballerie, X. Single-stranded positive-sense RNA viruses generated in days using infectious subgenomic amplicons. J. Gen. Virol. 2014, 95, 2462–2467, doi:10.1099/vir.0.068023-0.
  4. Fan, F.; Wood, K. V. Bioluminescent assays for high-throughput screening. Assay Drug Dev. Technol. 2007, 5, 127–136.

Reviewer 2 Report

Fernandes et. al. present a review and update on the use of replicons to develop antivirals against +strand RNA viruses.  It is timely given the current situation.  It is laid out well and covers all or most of the important RNA viruses.  Overall, it is a bit dense in places because they mention a lot of drugs and there is inconsistency in how they mention the compounds.  Some times they mention actual compounds and for others they mention the class or major structural moiety.  It would help to have some consistency here.  They show only a few structures at the end of the review and the criteria for including these and excluding others was not clear.  Also, they mention compounds as having activity at very high EC50 values and in most development programs these would be tossed so why they are mentioned is not clear.  Finally there are a number of English usage errors that need to be fixed.  Some examples:

Lines 33 and 34  The sentence should read ...most of these infections remain without...

Line 45 should read... protein, the level of activity of which, would reflect....

Line 115 should read ...efficiently replicating the viral....

Line 117 should read ...A BHK-21 cell line...

Line 118 change molecules to molecule

Line 122 change to ...with a quinoline scaffold...

Line 176 Delete Paving the way of DAAs development and start the sentence with It is worthy to note...

Line 188 should read Despite being discontinued...

Line 210 should read ;..benefited from HCV replicon....

So the manuscript needs to be checked for this and corrected.

Author Response

Point-by-point Response to Reviewer 2 Comments

Point 1: Fernandes et. al. presents a review and update on the use of replicons to develop antivirals against +strand RNA viruses.  It is timely given the current situation.  It is laid out well and covers all or most of the important RNA viruses.  Overall, it is a bit dense in places because they mention a lot of drugs and there is inconsistency in how they mention the compounds.  Sometimes they mention actual compounds and for others they mention the class or major structural moiety.  It would help to have some consistency here. 

Response 1: We thank the reviewer for the observation. To create a consistency, we standardized the compounds mention in the manuscript by the chemical class or biological functions for those which the chemical class was not found in the literature.

Point 2: They show only a few structures at the end of the review and the criteria for including these and excluding others was not clear. 

Response 2: The reviewer is right, the criteria for including molecules in Figure 4 was not clear. Thus, we changed the figure to show 2D structures of key compounds approved as antivirals against (+) ssRNA viruses.

Point 3: Also, they mention compounds as having activity at very high EC50 values and in most development programs these would be tossed so why they are mentioned is not clear.

Response 3: The authors agree with the reviewer that many compounds in the manuscript display very high EC50 values and would be tossed in most development programs. However, we intended to include in the manuscript all or most of the molecules that have already been tested using replicon-based assays, therefore, those compounds showing high EC50 values are cited in the text.

Point 4: Finally, there are a number of English usage errors that need to be fixed.  Some examples:

Lines 33 and 34 The sentence should read ...most of these infections remain without...

Response: The sentence in line 34 was corrected, according to reviewer’s suggestion.

Line 45 should read... protein, the level of activity of which, would reflect....

Response: The sentence in line 74 was corrected, according to reviewer’s suggestion.

Line 115 should read ...efficiently replicating the viral....

Response: The sentence in line 173 was corrected, according to reviewer’s suggestion.

Line 117 should read ...A BHK-21 cell line...

Response: The sentence in line 175 was corrected, according to reviewer’s suggestion.

Line 118 change molecules to molecule

Response: The sentence in line 176 was corrected, according to reviewer’s suggestion.

Line 122 change to ...with a quinoline scaffold...

Response: The sentence in line 181 was corrected, according to reviewer’s suggestion.

Line 176 Delete Paving the way of DAAs development and start the sentence with It is worthy to note...

Response: The sentence in line 235 was modified, according to reviewer’s suggestion.

Line 188 should read Despite being discontinued...

Response: The sentence in line 247 was corrected, according to reviewer’s suggestion.

Line 210 should read...benefited from HCV replicon....

Response: The sentence in line 260 was corrected, according to reviewer’s suggestion.

So, the manuscript needs to be checked for this and corrected.

Response: The manuscript was checked for English errors and corrected, as suggested by the reviewer.

Reviewer 3 Report

Single-stranded positive RNA viruses include several important human pathogens. Many of them are classified as BSL-3 agents, which limit the discovery of new inhibitors because the BLS-3 facilities are required to manipulate viruses. Replicons are a self-replicative viral subgenomic RNAs without infectious particles productions, which mimic the replication of viral genomic synthesis. Replicon could be used for high throughput assay to identify DAA compounds without biosafety concerns.

The authors review the use of (+) ssRNA viruses replicon systems for the discovery of antiviral agents and highlight some of the most promising identified inhibitors. Overall, the review is quite interesting to the field. Here are some concerns needed to be addressed.

  1. The authors mentioned the advantage of replicon system for drug discovery. It will be great if some disadvantage of replicon is discussed at the same time.
  2. Line 89-95. Flaviviridae includes genus members (Flavivirus, Hepacivirus, Pestivirus and Pegivirus). Traditionally, the nomenclature of structural and nonstructural proteins from different genus is different. I would like authors to correct the statement.

Author Response

Point-by-point Response to Reviewer 3 Comments

Single-stranded positive RNA viruses include several important human pathogens. Many of them are classified as BSL-3 agents, which limit the discovery of new inhibitors because the BLS-3 facilities are required to manipulate viruses. Replicons are a self-replicative viral subgenomic RNAs without infectious particles productions, which mimic the replication of viral genomic synthesis. Replicon could be used for high throughput assay to identify DAA compounds without biosafety concerns.

The authors review the use of (+) ssRNA viruses replicon systems for the discovery of antiviral agents and highlight some of the most promising identified inhibitors. Overall, the review is quite interesting to the field. Here are some concerns needed to be addressed.

Point 1: The authors mentioned the advantage of replicon system for drug discovery. It will be great if some disadvantage of replicon is discussed at the same time.

Response 1: The authors thank the reviewer for the suggestion. A disadvantage of replicons also an alternative approach to those systems were added to the manuscript (lines 82 to 89).

Point 2: Line 89-95. Flaviviridae includes genus members (Flavivirus, Hepacivirus, Pestivirus and Pegivirus). Traditionally, the nomenclature of structural and nonstructural proteins from different genus is different. I would like authors to correct the statement.

Response 2: We thank the reviewer for the observation. Lines 143 to 151 were modified to: “The Flaviviridae is a large family of spherical-shaped viruses of ~ 40-60 nm in diameter, divided into four genera: Flavivirus, Hepacivirus, Pestivirus and Pegivirus. The Flavivirus and Hepacivirus genera comprise several widely studied species, since they cause human diseases. The Flavivirus genome of ~ 11 kb encodes a single polyprotein that is co- and post-translationally processed into three structural proteins: capsid (C), pre-membrane (prM), and envelope (E); and, seven nonstructural (NS) proteins NS1, NS2A, NS2B, NS3, NS4A, NS4B, and NS5 (Figure 2a). Similarly, a single polyprotein is encoded by Hepacivirus genome and cleaved into four structural proteins: core (C), envelope proteins 1 (E1) and 2 (E2), and polypeptide 7 (p7), plus six NS proteins: NS2, NS3, NS4A, NS4B, NS5A and NS5B”.

Round 2

Reviewer 1 Report

I am satisfied with the changes made to the manuscript.